# BOOD: Boundary-based Out-Of-Distribution Data Generation

Qilin Liao [1]  Shuo Yang [2] [*]  Bo Zhao [3]  Ping Luo [1]  Hengshuang Zhao [1] [*]

## Abstract

Harnessing the power of diffusion models to synthesize auxiliary training data based on *latent space* features has proven effective in enhancing out-of-distribution (OOD) detection performance. However, extracting effective features outside the in-distribution (ID) boundary in *latent space* remains challenging due to the difficulty of identifying decision boundaries between classes. This paper proposes a novel framework called Boundary-based Out-Of-Distribution data generation (BOOD), which synthesizes high-quality OOD features and generates human-compatible outlier images using diffusion models. BOOD first learns a text-conditioned latent feature space from the ID dataset, selects ID features closest to the decision boundary, and perturbs them to cross the decision boundary to form OOD features. These synthetic OOD features are then decoded into images in pixel space by a diffusion model. Compared to previous works, BOOD provides a more training efficient strategy for synthesizing informative OOD features, facilitating clearer distinctions between ID and OOD data. Extensive experimental results on common benchmarks demonstrate that BOOD surpasses the state-of-the-art method significantly, achieving a 29.64% decrease in average FPR95 (40.31% vs. 10.67%) and a 7.27% improvement in average AUROC (90.15% vs. 97.42%) on the CIFAR-100 dataset.

## 1. Introduction

In the field of open-world learning, machine learning models will encounter various inputs from unseen classes, thus be confused and make untrustworthy predictions. Out-Of-Distribution (OOD) detection, which flags outliers during training, is a non-trivial solution for helping models form a boundary around the ID (in-distribution) data (Du et al., 2023). Recent works have shown that training neural networks with auxiliary outlier datasets is promising for helping the model to form a decision boundary between ID and OOD data (Hendrycks et al., 2019; Liu et al., 2020; Katz-Samuels et al., 2022; Ming et al., 2022). However, the process of manually preparing OOD data for model training incurs substantial costs, both in terms of human resources investment and time consumption. Additionally, it's impossible to collect data distributed outside the data distribution boundary, which can not be captured in the real world as shown in Figure 1.

To address this problem, recent works have demonstrated pipelines regarding automating OOD data generation, which significantly decreases the labor intensity during creating auxiliary datasets (Du et al., 2022; Tao et al., 2023a; Du et al., 2023; Chen et al., 2024). As a representative of them, DreamOOD (Du et al., 2023) models the training data distribution and samples visual embeddings from low-likelihood regions as OOD auxiliary data in a text-conditioned *latent space*, then decoding them into images through a diffusion model. However, due to the lack of an explicit relationship between the low-likelihood regions and the decision boundaries between classes, the DreamOOD (Du et al., 2023) can **not** guarantee the generated images always lie on the decision boundaries, which have demonstrated efficacy in enhancing the robustness of the ID classifier and refining its decision boundaries (Ming et al., 2022). Chen et al. (2020) proposes OOD detection frameworks targeting the OOD boundary by searching for a similarity threshold between class center features and encoded input features. Pei et al. (2022) harnesses GAN (Goodfellow et al., 2014) to train a boundary-aware discriminator and an OOD generator for distinguishing ID and OOD data. However, these two methods still lack the ability to find an explicit distribution boundary between ID and OOD area, and generate **image-level** OOD dataset.

In this paper, we introduce a new framework, BOOD (Boundary-based Out-Of-Distribution data generation), which explicitly enables us to generate images located around decision boundaries between classes, thus providing high-quality and informative features for OOD detection.

---

[1] The University of Hong Kong, Hong Kong, China [2] Department of Computer Science, Harbin Institute of Technology (Shenzhen), Shenzhen, China [3] School of AI, Shanghai Jiao Tong University, Shanghai, China. Correspondence to: Shuo Yang <shuoyang@hit.edu.cn>, Hengshuang Zhao <hszhao@cs.hku.hk>.

*Proceedings of the 42nd International Conference on Machine Learning*, Vancouver, Canada. PMLR 267, 2025. Copyright 2025 by the author(s).

Bear                                    Tiger

ID

OOD

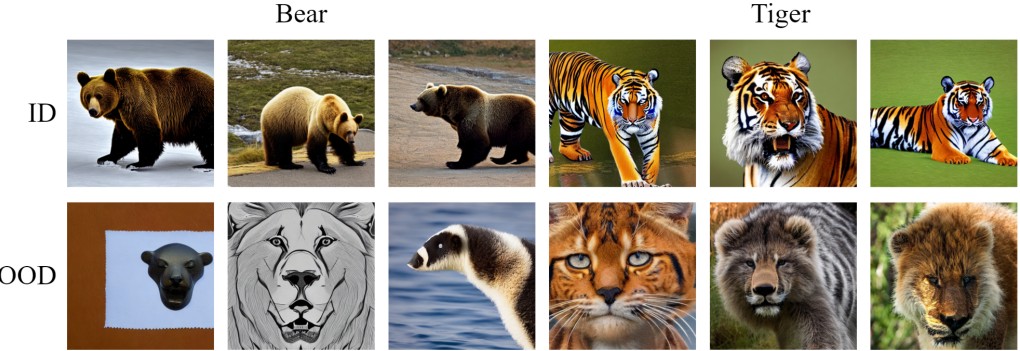

*Figure 1.* **Top**: images generated from ID features. **Bottom**: images generated from OOD features. Compared to preparing ID image datasets, preparing OOD image datasets incurs substantial costs in terms of resource allocation, particularly with respect to labor and time investment. Moreover, certain OOD images, as illustrated in the above figure, are impossible to acquire through real-world data collection methods. Consequently, there exists a pressing need for the development of automated pipelines capable of generating OOD datasets.

The challenging part lies in the following: *(1) Identifying the data distribution boundary accurately*, and *(2) Synthesizing the informative outlier features based on the identified data distribution boundaries.* Our innovative framework addresses the aforementioned challenges by: *(1) an adversarial perturbation strategy, which successfully identifies the features closest to the decision boundary by calculating the minimal perturbation steps imposed on the feature to change the model's prediction,* and *(2) an outlier feature synthesis strategy, which generates the outlier features by perturbing the identified boundary ID features along with the gradient ascent direction.* The synthetic outlier features are subsequently fed into a diffusion model to generate the OOD images. To guarantee the synthetic feature space is compatible with the diffusion-model-input-space (class token embedding space), we employ a class embedding alignment strategy during the image encoder training following Du et al. (2023).

Before delving into details, we summarize our contributions as below:

- To our best knowledge, BOOD is the first framework that enables generating image-level OOD data lying around the decision boundaries explicitly, thus providing informative features for shaping the decision boundaries between ID and OOD data.

- We propose two key methodologies to address the challenges in synthesizing the OOD features: (1) Identifying the ID boundary data by counting their minimum perturbation steps to cross the decision boundaries for all ID features. (2) Synthesizing the informative OOD features lying around the decision boundaries by perturbing the ID boundary features towards the gradient ascent direction.

- Our method demonstrates superior performance

improvement across two challenging benchmarks, achieving state-of-the-art results on CIFAR-100 and IMAGENET-100 datasets. For instance, on CIFAR-100, BOOD improves the average performance on detecting 5 OOD datasets from 40.31% to 10.67% in FPR95 and from 90.15% to 97.42% in AUROC. Moreover, we conducted extensive quantitative ablation analyses to provide a deeper insight into BOOD's efficiency mechanism.

## 2. Preliminaries

**Latent space formation.** Given an ID training dataset, $\mathcal{D}_{\text{id}} = \{(x_i, y_i)\}_{i=1}^{m}$, where $x_i \in \mathcal{X}$ and $y_i \in \mathcal{Y}$. $\mathcal{X}$ denotes the input space and $\mathcal{Y} \in \{1, 2, ...V\}$ denotes the label space. Let $h_\theta(x) : \mathcal{X} \rightarrow \mathbb{R}^n$ denote the image feature encoder, where $\mathbb{R}^n$ denotes the feature space. The output of $h_\theta$ is supposed to be an $n$-dimensional vector representing the encoded image feature. We denote $f(x) = CosSim(h_\theta(x), \Gamma(y))$ as the cosine image classifier, whose output is assumed to be a $v$-dimensional vector that performs as a discrete probability function representing prediction probability for each class. $\Gamma(y)$ represents the class token embedding encoded by feeding class name $y$ into CLIP (Radford et al., 2021) text encoder.

**OOD detection.** In real-world applications of machine learning models, a reliable classification system must exhibit dual capabilities: it should categorize familiar in-distribution (ID) samples, and it is able to recognize and flag out-of-distribution (OOD) inputs that belong to unknown classes not represented in the original training set $y \notin \mathcal{Y}$. Thus, having an OOD detector can solve this problem. OOD detection can be formulated as a binary classification problem (Ming et al., 2022), and the goal is to decide whether an input is from ID or OOD. We denote the OOD detection as $g_\theta(x) : \mathcal{X} \rightarrow \{ID, OOD\}$ mathematically.

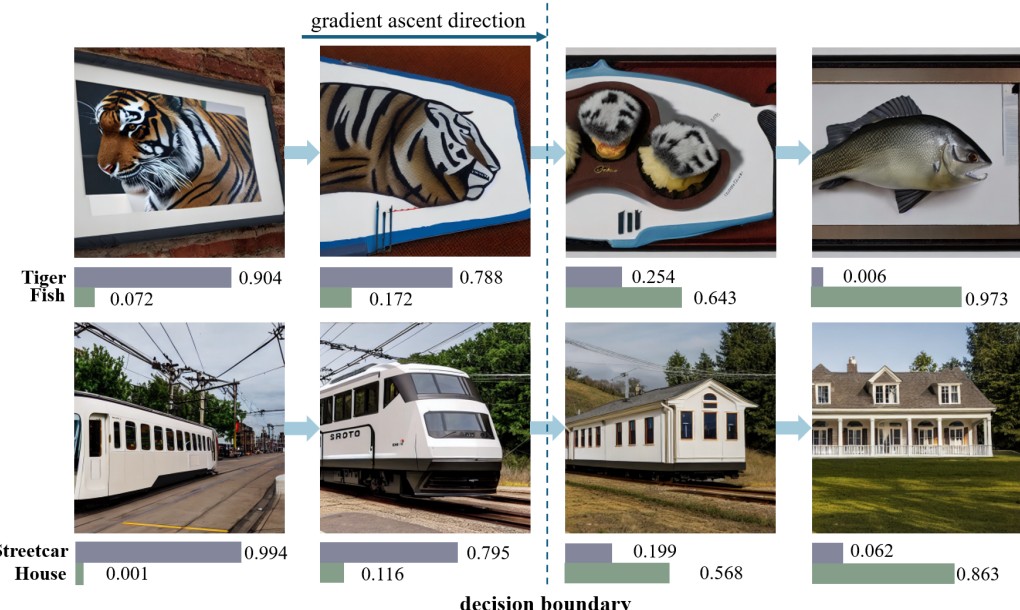

*Figure 2.* Illustration of perturbing ID boundary feature process. The bar charts under each image represent the prediction probability of the perturbed features by the image classifier. After each perturbation, the prediction probability of the original class decreases. When the prediction of the image classifier switches, we consider the obtained feature crossed the decision boundary.

**Diffusion-based image generation.** Diffusion models demonstrate formidable prowess in generating authentic and lifelike content. Their robust capabilities extend to various applications, with particular efficacy in tasks such as the creation of synthetic images. We can synthesize images in a specific distribution by conditioning on class labels or text descriptions (Ramesh et al., 2022). Stable Diffusion (Rombach et al., 2022) is a text-to-image model which enables generating particular images conditioned by text prompts. For a given class name $y$, the generating process can be denoted by:

$$x \sim P(x|Z_y) \qquad (1)$$

where $Z_y = \Gamma(Y)$ denotes a specific textual representation of class label $y$ with prompting, and we denote the whole prompting as $Y$. For instance, $Y =$ "A picture of $[y]$". $\Gamma$ denotes the CLIP (Radford et al., 2021) model's text encoder.

## 3. BOOD: Boundary-based Out-Of-Distribution data generation

Images situated near the decision boundary offer informative OOD insights, which can significantly enhance the ability of OOD detection models to establish accurate boundaries between ID and OOD data, thereby improving overall detection performance. In this paper, we propose a framework BOOD (Boundary-based Out-Of-Distribution data generation), which enables us to generate human-compatible synthetic images decoding from *latent space* features lying around the decision boundaries among ID classes. The challenging part lies in identifying the ID boundary features and synthesizing outlier features located around the decision boundary, which have demonstrated efficacy in enhancing the robustness of the ID classifier and refining its decision boundaries (Ming et al., 2022).

### 3.1. Building the Text-Conditioned Latent Space

Aiming at ensuring the image features are suitable for being decoded by the diffusion model, we first create an image feature space that is aligned with the diffusion-model-input space. To achieve this, we train the image encoder $h_\theta$ by aligning the extracted image features $h_\theta(x)$ with their corresponding class token embeddings $\Gamma(y)$, which match the input space with the diffusion model. The resulting generated features form a text-conditioned *latent space*. Following DreamOOD (Du et al., 2023), we train the image encoder $h_\theta$ with the following loss function:

$$\mathcal{L}_c = \mathbb{E}_{(x,y) \sim \mathcal{D}_{\text{id}}}[-log \frac{exp(\Gamma(y)^\top z/t)}{\Sigma_{j=1}^{C} exp(\Gamma(y_j)^\top z/t)}] \qquad (2)$$

where $z = h_\theta(x)/\|(x)\|_2$ is the $L_2$-normalized image feature embedding, $t$ is the temperature, $h_\theta$ denotes the text-conditioned image feature encoder, and $\Gamma(y)$ denotes the class token embedding encoded by feeding class name $y$ into CLIP (Radford et al., 2021) text encoder. After training

*Table 1.* OOD detection results for CIFAR-100 as the in-distribution data. We report standard deviations estimated across 3 runs. Bold numbers are superior results, and the last row is the improvement of our method over previous state-of-the-art DreamOOD (Du et al., 2023).

| Methods | OOD Datasets | | | | | | | | | | | | ID ACC |
| | SVHN | | PLACES365 | | LSUN | | ISUN | | TEXTURES | | Average | | |
| | FPR95↓ | AUROC↑ | FPR95↓ | AUROC↑ | FPR95↓ | AUROC↑ | FPR95↓ | AUROC↑ | FPR95↓ | AUROC↑ | FPR95↓ | AUROC↑ | |
|---|---|---|---|---|---|---|---|---|---|---|---|---|---|
| MSP (Hendrycks & Gimpel, 2017) | 87.35 | 69.08 | 81.65 | 76.71 | 76.40 | 80.12 | 76.00 | 78.90 | 79.35 | 77.43 | 80.15 | 76.45 | 79.04 |
| ODIN (Liang et al., 2018) | 90.95 | 64.36 | 79.30 | 74.87 | 75.60 | 78.04 | 53.10 | 87.40 | 72.60 | 79.82 | 74.31 | 76.90 | 79.04 |
| Mahalanobis (Lee et al., 2018b) | 87.80 | 69.98 | 76.00 | 77.90 | 56.80 | 85.83 | 59.20 | 86.46 | 62.45 | 84.43 | 68.45 | 80.92 | 79.04 |
| Energy (Liu et al., 2020) | 84.90 | 70.90 | 82.05 | 76.00 | 81.75 | 78.36 | 73.55 | 81.20 | 78.70 | 78.87 | 80.19 | 77.07 | 79.04 |
| GODIN (Hsu et al., 2020) | 63.95 | 88.98 | 80.65 | 77.19 | 60.65 | 88.36 | 51.60 | 92.07 | 71.75 | 85.02 | 65.72 | 86.32 | 76.34 |
| KNN (Sun et al., 2022) | 81.12 | 73.65 | 79.62 | 78.21 | 63.29 | 85.56 | 73.92 | 79.77 | 73.29 | 80.35 | 74.25 | 79.51 | 79.04 |
| ViM (Wang et al., 2022) | 81.20 | 77.24 | 79.20 | 77.81 | 43.10 | 90.43 | 74.55 | 83.02 | 61.85 | 85.57 | 67.98 | 82.81 | 79.04 |
| ReAct (Sun et al., 2021) | 82.85 | 70.12 | 81.75 | 76.25 | 80.70 | 83.03 | 67.40 | 83.28 | 74.60 | 81.61 | 77.46 | 78.86 | 79.04 |
| DICE (Sun & Li, 2022) | 83.55 | 72.49 | 85.05 | 75.92 | 94.05 | 73.59 | 75.20 | 80.90 | 79.80 | 77.83 | 83.53 | 76.15 | 79.04 |
| *Synthesis-based methods* | | | | | | | | | | | | | |
| GAN (Lee et al., 2018b) | 89.45 | 66.95 | 88.75 | 66.76 | 82.35 | 75.87 | 83.45 | 73.49 | 92.80 | 62.99 | 87.36 | 69.21 | 70.12 |
| VOS (Du et al., 2022) | 78.50 | 73.11 | 84.55 | 75.85 | 59.05 | 85.72 | 72.45 | 82.66 | 75.35 | 80.08 | 73.98 | 79.48 | 78.56 |
| NPOS (Tao et al., 2023a) | 11.14 | 97.84 | 79.08 | 71.30 | 56.27 | 82.43 | 51.72 | 85.48 | 35.20 | 92.44 | 46.68 | 85.90 | 78.23 |
| DreamOOD (Du et al., 2023) | 58.75 | 87.01 | 70.85 | 79.94 | 24.25 | 95.23 | 1.10 | 99.73 | 46.60 | 88.82 | 40.31 | 90.15 | 78.94 |
| **BOOD** | **5.42** ±0.5 | **98.43**±0.1 | **40.55**±1 | **90.76**±0.5 | **2.06**±0.8 | **99.25**±0.1 | **0.22**±0.15 | **99.91**±0.02 | **5.1**±1 | **98.74**±0.2 | **10.67**±0.95 | **97.42**±0.1 | 78.03 ±0.1 |
| Δ **(improvements)** | **+53.33** | **+11.42** | **+30.3** | **+10.82** | **+22.19** | **+4.02** | **+0.88** | **+0.18** | **+41.5** | **+9.92** | **+29.64** | **+7.27** | |

the image encoder $h_\theta$, the image classifier $f$ can be simply formulated as a cosine classifier between the encoded image features $h_\theta(x)$ and the class token embeddings $\Gamma(y)$.

## 3.2. Synthesizing OOD features and Generating images

---

**Algorithm 1** BOOD: Boundary-based Out-Of-Distribution data generation

---

**Input:** In-distribution training data $\mathcal{D}_{id} = \{(x_i, y_i)\}_{i=1}^m$, initial model parameters $\theta$ for learning the text-conditioned *latent space*, diffusion model.
**Output:** Synthetic images $x_{ood}$.
// Section. 3.1: Building the text-conditioned latent space
1. Extract token embeddings $\Gamma(y)$ of the ID label $y \in \mathcal{Y}$.
2. Learn the text-conditioned latent representation space by Equation 2.
// Section. 3.2: Synthesizing OOD features and generating images
1. Calculate the distances for each feature and select the ID boundary features with Equation 3.
2. Perturb the selected ID boundary features to cross the decision boundary with Equation 4 and Equation 5.
3. Decode the outlier embeddings into the pixel-space OOD images via diffusion model by Equation 6.

---

After obtaining a well-established text-conditioned image feature *latent space*, our framework proposes the generation of outlier images through a three-step process. Firstly, we estimate each feature's distance to the decision boundary by counting their perturbation steps to cross the decision boundaries, and select the ID boundary features by choosing those features with minimal distances in Sec. 3.2.1. Subsequently, we push the identified ID boundary features to the location around the decision boundary to synthesize OOD features by perturbing them along with the gradient ascent direction until the model's prediction switches in

Sec. 3.2.2. We finally decode the synthesized OOD features through the diffusion model and generate OOD images in Sec. 3.2.3. Figure 4 is a visual representation of Sec. 3.2.1 and Sec. 3.2.2.

### 3.2.1. BOUNDARY FEATURE IDENTIFICATION

We believe that the ID features distributed near the decision boundary are more sensitive to perturbation, as slight perturbation can push them across the decision boundary, making them ideal candidates for synthesizing OOD features in Sec. 3.2.2. Thus, the target at this stage is to select features that are closest to the decision boundary. Introduced by Chakraborty et al. (2018) and Kurakin et al. (2017), adversarial attack endeavors to perturb a data point to the smallest possible extent to cross the model's decision boundary. Inspired by Yang et al. (2024b), Mickisch et al. (2020) and Yousefzadeh & O'Leary (2020), our objective is to determine the minimal distance required for an in-distribution (ID) feature to traverse the decision boundary. This is accomplished by quantifying the number of steps, denoted as $k$, necessary to perturb the ID feature along the gradient ascent direction until it changes the model's prediction.

Below is the working principle for a given feature $(z, y)$:

$$z_{adv}^{(k+1)} = z_{adv}^{(k)} + \alpha \cdot sign(\nabla_{z_{adv}^{(k)}} l(f_\theta(z_{adv}^{(k)}), y)), k \in [0, K] \quad (3)$$

where $\alpha$ denotes the step size of a single perturbation, $z_{adv}^{(k)}$ denotes adversarial feature at step $k$, $l$ is the loss function, $f_\theta$ denotes the image classifier and $K$ denotes the maximum iteration. The process keeps iterating until $f_\theta \neq y$ or $k = K$, indicating that the adversarial feature $z_{adv}^{(k)}$ has crossed the decision boundary or $k$ exceeds the maximum allowed iteration number $K$. We provide visualization of this process

*Table 2.* OOD detection results for IMAGENET-100 as the in-distribution data. We report standard deviations estimated across 3 runs. Bold numbers are superior results, and the last row is the improvement of our method over previous state-of-the-art DreamOOD (Du et al., 2023).

| Methods | OOD Datasets | | | | | | | | | | ID ACC |
|---|---|---|---|---|---|---|---|---|---|---|---|
| | iNaturalist | | Places | | Sun | | Textures | | Average | | |
| | FPR95↓ | AUROC↑ | FPR95↓ | AUROC↑ | FPR95↓ | AUROC↑ | FPR95↓ | AUROC↑ | FPR95↓ | AUROC↑ | |
| MSP (Hendrycks & Gimpel, 2017) | 31.80 | 94.98 | 47.10 | 90.84 | 47.60 | 90.86 | 65.80 | 83.34 | 48.08 | 90.01 | 87.64 |
| ODIN (Liang et al., 2018) | 24.40 | 95.92 | 50.30 | 90.20 | 44.90 | 91.55 | 61.00 | 81.37 | 45.15 | 89.76 | 87.64 |
| Mahalanobis (Lee et al., 2018b) | 91.60 | 75.16 | 96.70 | 60.87 | 97.40 | 62.23 | 36.50 | 91.43 | 80.55 | 72.42 | 87.64 |
| Energy (Liu et al., 2020) | 32.50 | 94.82 | 50.80 | 90.76 | 47.60 | 91.71 | 63.80 | 80.54 | 48.68 | 89.46 | 87.64 |
| GODIN (Hsu et al., 2020) | 39.90 | 93.94 | 59.70 | 89.20 | 58.70 | 90.65 | 39.90 | 92.71 | 49.55 | 91.62 | 87.38 |
| KNN (Sun et al., 2022) | 28.67 | 95.57 | 65.83 | 88.72 | 58.08 | 90.17 | 12.92 | 90.37 | 41.38 | 91.20 | 87.64 |
| ViM (Wang et al., 2022) | 75.50 | 87.18 | 88.30 | 81.25 | 88.70 | 81.37 | 15.60 | 96.63 | 67.03 | 86.61 | 87.64 |
| ReAct (Sun et al., 2021) | 22.40 | 96.05 | 45.10 | 92.28 | 37.90 | 93.04 | 59.30 | 85.19 | 41.17 | 91.64 | 87.64 |
| DICE (Sun & Li, 2022) | 37.30 | 92.51 | 53.80 | 87.75 | 45.60 | 89.21 | 50.00 | 83.27 | 46.67 | 88.19 | 87.64 |
| *Synthesis-based methods* | | | | | | | | | | | |
| GAN (Lee et al., 2018a) | 83.10 | 71.35 | 83.20 | 69.85 | 84.40 | 67.56 | 91.00 | 59.16 | 85.42 | 66.98 | 79.52 |
| VOS (Du et al., 2022) | 43.00 | 93.77 | 47.60 | 91.77 | 39.40 | 93.17 | 66.10 | 81.42 | 49.02 | 90.03 | 87.50 |
| NPOS (Tao et al., 2023a) | 53.84 | 86.52 | 59.66 | 83.50 | 53.54 | 87.99 | **8.98** | **98.13** | 44.00 | 89.04 | 85.37 |
| DreamOOD (Du et al., 2023) | 24.10 | 96.10 | 39.87 | 93.11 | **36.88** | 93.31 | 53.99 | 85.56 | 38.76 | 92.02 | 87.54 |
| **BOOD** | **18.33**±0.3 | **96.74**±0.2 | **33.33**±0.5 | **94.08**±0.4 | 37.92±0.2 | **93.52**±0.1 | 51.88±0.5 | 85.41±0.5 | **35.37**±0.3 | **92.44**±0.1 | 87.92±0.05 |
| Δ (improvements) | +5.77 | +0.64 | +6.54 | +0.97 | -1.04 | +0.21 | +2.11 | -0.15 | +3.39 | +0.42 | |

in Figure 2.

During each iteration, our method perturbs the adversarial feature in a direction that maximizes the change in the model's prediction. The minimum number of iterations $k$ necessary to create an adversarial example $z_{adv}$ from a given feature $z$ that crosses the decision boundary, can be employed as a proxy for the shortest distance between that data point and the decision boundary. This relationship is expressed as $d(z, y) = k$, where $k$ is bounded by $[0, K]$. Thus, we can obtain the distance set for all ID features to the decision boundaries $\mathcal{D}$ and select the ID boundary features that have minimal distances to the decision boundary, denoted as $z_{id} \in \{z | d(z, y) \in \mathcal{D}_{r\%}\}$ where $\mathcal{D}_{r\%}$ denotes the smallest $r\%$ of distance set $\mathcal{D}$ and $r$ denotes the selection ratio of ID boundary selection.

### 3.2.2. OOD FEATURE SYNTHESIZING

The features distributed around the decision boundary can provide high-quality OOD information to facilitate the OOD detection model to form precise ID-OOD boundaries. So we aim to perturb the selected ID boundary features $z_{id}$ to the location around the decision boundary, where we might synthesize informative features. These OOD features, denoted as $z_{ood}$, will be decoded into outlier images that are distributed around the OOD detection boundary. We summarize the perturbation process in the following module:

$$
\begin{array}{l}
\textbf{While } (f(z_{id}) = y) \textbf{ do} \\
\quad z_{id} = z_{id} + \alpha \cdot sign\left(\nabla_{z_{id}} l(f_\theta(z_{id}), y)\right) \quad (4) \\
\textbf{end} \\
\qquad\qquad z_{ood} = z_{id} \\
\textbf{for } i \leftarrow 0 \textbf{ to } c \\
\quad z_{ood}^{(i+1)} = z_{ood}^{(i)} + \alpha \cdot sign\left(\nabla_{z_{ood}^{(i)}} l(f_\theta(z_{ood}^{(i)}), y)\right) \quad (5) \\
\textbf{end}
\end{array}
$$

Consider a selected ID boundary feature $z_{id}$, we perturb it following the direction of gradient ascent until the prediction of the image classifier $f_\theta$ switches ($f(z_{id}) \neq y$). We continue perturbing it for $c$ steps to guarantee it is adequately distant from the ID boundary. We provide ablation studies on $\alpha$ and $c$ in Sec. 4.3.2.

### 3.2.3. OOD IMAGE GENERATION

To generate the outlier images, we finally decode the synthetic OOD feature embeddings $z_{ood}$ through a diffusion model. Following Du et al. (2023), we replace the origin token embedding $\Gamma(y)$ in the textual representation $Z_y$ with our synthetic OOD embedding $z_{ood}$. The generation process can be formulated as:

$$
x_{ood} \sim P(x | Z_{ood}) \tag{6}
$$

where $x_{ood}$ denotes the synthetic OOD images and $Z_{ood}$ denotes the textual representation $Z_y$ with $\Gamma(y)$ replaced by $z_{ood}$. We summarize our methodology in Algorithm 1.

### 3.3. Regularizing OOD detection model

After synthesizing the OOD images, we regularize the OOD classification model with the following loss function:

$$
\begin{aligned}
\mathcal{L}_{OOD} = \mathbb{E}_{x_{id} \sim \mathcal{D}_{id}} & \left[ -\log \frac{\exp(\phi(E(g_\theta(x_{id}))))}{1 + \exp(\phi(E(g_\theta(x_{id}))))} \right] \\
+ \mathbb{E}_{x_{ood} \sim \mathcal{D}_{ood}} & \left[ -\log \frac{1}{1 + \exp(\phi(E(g_\theta(x_{ood}))))} \right]
\end{aligned} \tag{7}
$$

where $\phi$ denotes a 3-layer MLP function of the same structure as VOS (Du et al., 2022), $E$ denotes the energy function

steps crossing boundary

-1      0      1      2

step size

0      0.045      0.09      0.135

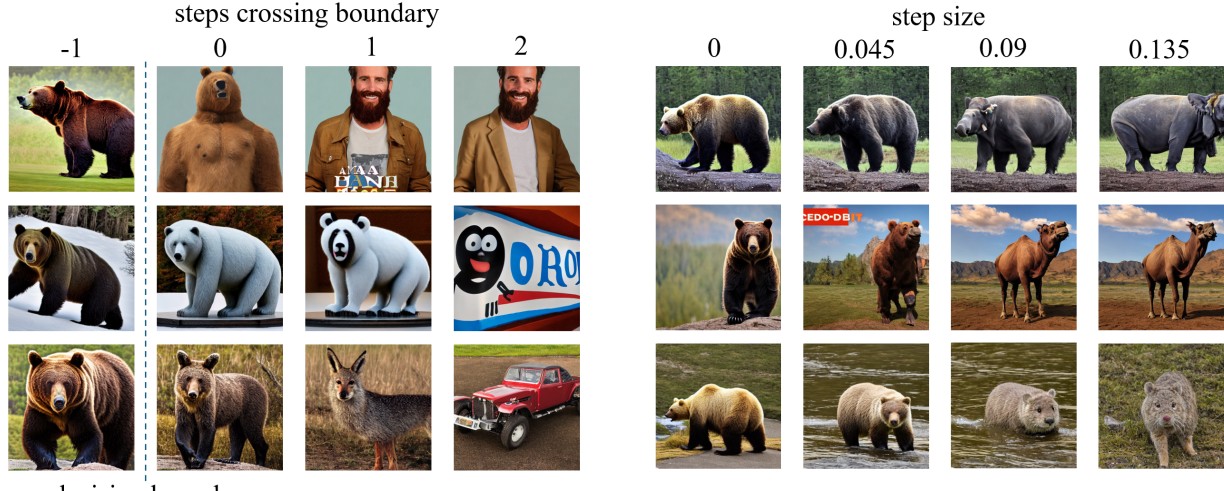

decision boundary

*Figure 3.* **Left**: the effect of perturbing steps $c$ after crossing the boundary, **Right**: the effect of step size $\alpha$.

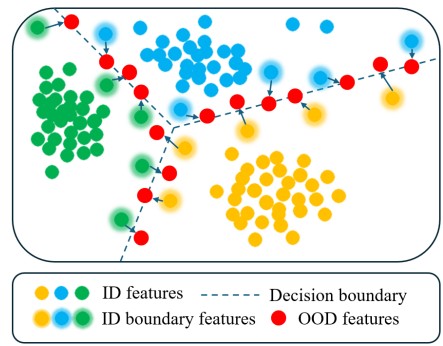

| | ID features | - - - - - | Decision boundary |
| | ID boundary features | ● | OOD features |

*Figure 4.* Illustration of the identified ID boundary features and perturbing them to cross the decision boundary.

and $g_\theta$ denotes the output of OOD classification model. The final training objective function combines cross-entropy loss and OOD regularization loss, which can be reflected by $\mathcal{L}_{CE} + \beta \cdot \mathcal{L}_{OOD}$, where $\beta$ denotes the weight of the OOD regularization.

## 4. Experiments and Analysis

### 4.1. Experimental setup and Implementation details

**Datasets.** Following DreamOOD (Du et al., 2023), we select CIFAR-100 and IMAGENET-100 (Deng et al., 2009) as ID image datasets. As the OOD datasets should not overlap with ID datasets, we choose SVHN (Netzer et al., 2011), PLACES365 (Zhou et al., 2018), TEXTURES(Cimpoi et al., 2014), LSUN (Yu et al., 2015), ISUN (Xu et al., 2015) as OOD testing image datasets for CIFAR-100. For IMAGENET-100, we choose INATURALIST (Horn et al., 2018), SUN (Xiao et al., 2010), PLACES (Zhou et al.,

2018) and TEXTURES (Cimpoi et al., 2014), following MOS (Huang & Li, 2021).

**Training details.** The ResNet-34 (He et al., 2016) architecture was employed as the training network for both the CIFAR-100 and IMAGENET-100 datasets. The initial learning rate was set to 0.1, with a cosine learning rate decay schedule implemented. A batch size of 160 was utilized. In the construction of the *latent space*, the temperature parameter $t$ was assigned a value of 1. In the boundary feature selection process, the initial pruning rate $r$ was established at 5, with an initial total step $K$ of 100. The step size $\alpha$ was configured to 0.015. The hyper parameters for the OOD feature synthesis step were maintained consistent with those of the boundary feature identification process. A total of 1000 images per class were generated using Stable Diffusion v1.4, yielding a comprehensive set of 100,000 OOD images. For the regularization of the OOD detection model, the $\beta$ parameter was set to 1.0 for IMAGENET-100 and 2.5 for CIFAR-100.

**Evaluation metrics.** We evaluate the performance using three key metrics: (1) the false positive rate at 95% true positive rate (FPR95) for OOD samples, (2) the area under the receiver operating characteristic curve (AUROC), and (3) in-distribution classification accuracy (ID ACC). These metrics collectively assess the model's discriminative capability, overall performance, and retention of in-distribution task proficiency.

### 4.2. Comparison with State-of-the-art

BOOD shows outstanding performance improvement compared to previous state-of-the-art methods. As shown in Table 1 and 2, we compare BOOD with other methods,

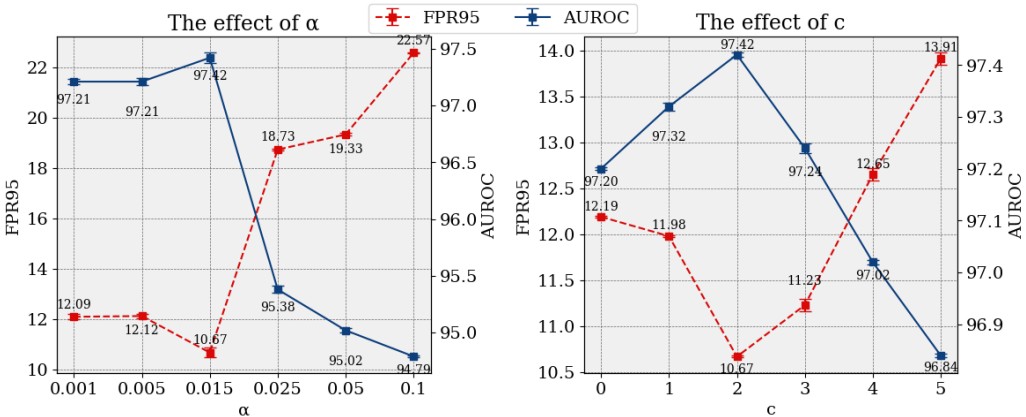

*Figure 5.* **Left**: The effect of step size $\alpha$. **Right**: The effect of perturbation steps $c$ after crossing the boundary.

*Table 3.* Ablation on OOD feature synthesizing methodologies

| Methods | | Criteria (Avg.) | | |
|---|---|---|---|---|
| boundary identification | feature perturbation | FPR95 ↓ | AUROC ↑ | ID ACC |
| ✓ | | 99.61 | 7.83 | 76.51 |
| | ✓ | 44.26 | 89.79 | 77.59 |
| ✓ | ✓ | **10.67** | **97.42** | **77.64** |

including Maximum Softmax Probability (Hendrycks & Gimpel, 2017), ODIN score (Liang et al., 2018), Mahalanobis score (Lee et al., 2018b), Energy score(Liu et al., 2020), Generalized ODIN (Hsu et al., 2020), KNN distance (Sun et al., 2022), VIM score (Wang et al., 2022), ReAct (Sun et al., 2021) and DICE (Sun & Li, 2022). Additionally, we compare BOOD with another four synthesis-based methods, including GAN-based synthesis (Lee et al., 2018b), VOS(Du et al., 2022), NPOS (Tao et al., 2023a) and DreamOOD (Du et al., 2023) as they have a closer relationship with us. BOOD surpasses the state-of-the-art method significantly, achieving a 29.64% decrease in average FPR95 (40.31% vs. 10.67%) and a 7.27% improvement in average AUROC (90.15% vs. 97.42%) on the CIFAR-100 dataset. BOOD's performance also surpasses the state-of-the-art methodologies on the IMAGENET-100 dataset.

## 4.3. Ablation Studies and Hyper-parameter Analysis

### 4.3.1. ABLATION ON OOD FEATURE SYNTHESIZING METHODOLOGIES

We ablate the effect of boundary identification and feature perturbation. As shown in Table 3, we conduct 3 experiments: (1) directly decode the ID features selected by Section 3.2.1, (2) randomly choose ID features and perturb them to the boundary (Section 3.2.2), (3) full BOOD. The results demonstrate that both the boundary feature identification and OOD feature perturbation modules are essential

for achieving the best result. ID boundary features are more sensitive to perturbation, which makes them optimal candidates for perturbation. The generated features lying around the decision boundary can provide high-quality OOD information to help the OOD detection model regularize the ID-OOD decision boundary.

### 4.3.2. HYPER PARAMETERS SENSITIVE ANALYSIS

**The effect of step size $\alpha$.** We show the effect of step size $\alpha$ in Figure 5 (left). Employing a smaller step size allows for minor perturbations of the instance $x$ in each iteration and facilitates a more nuanced differentiation between samples across different distances. It also guarantees that the perturbed features are in a more accurate direction towards the decision boundary. We choose the step size $\alpha$ as 0.015 in our experiments. Figure 3 (left) illustrates the effect of $\alpha$: when $\alpha$ increases, the discrepancy between iteration increases.

**The effect of perturbation steps $c$ after crossing the boundary.** We analyze the effect of perturbation steps $c$ after crossing the boundary in Figure 5 (right) to explore whether it will extract more efficient features. We vary steps $c \in \{0, 1, 2, 3, 4, 5\}$ and observe that when $c = 2$, BOOD shows the best performance. Employing a large $c$ may force the feature to step into the ID region, and choosing a small $c$ may not guarantee the perturbed feature is adequately distant from the ID boundaries. Figure 3 (right) shows the effect of $c$: as the number of steps crossing the boundary augment, the generated images gradually transform into another classes or distribute outside the distribution boundary.

**The effect of $r$.** We show the effect of pruning rate $r$ in Figure 6 (left). We vary rate $r \in \{2.5, 5, 10, 20\}$ and observe that BOOD shows best performance when we employ a moderate pruning rate. Insufficient pruning (small $r$) may limit the diversity of generated OOD images (not enough features), while excessive pruning (large $r$) risks selecting

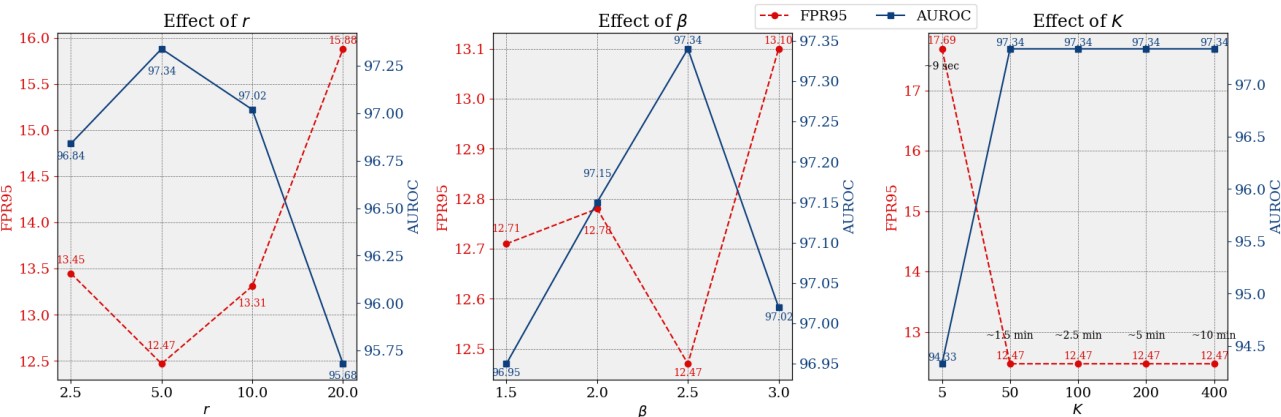

*Figure 6.* Left:The effect of pruning rate $r$. Middle: The effect of $\beta$. Right: The effect of maximum perturbation steps $K$

*Table 4.* Computational cost comparison

| Computational Cost | Building latent space | OOD features synthesizing | OOD image generation | OOD detection model regularization | Total |
|---|---|---|---|---|---|
| BOOD | ∼0.62h | ∼0.1h | ∼7.5h | ∼8.5h | ∼16.72h |
| DreamOOD | ∼0.61h | ∼0.05h | ∼7.5h | ∼8.5h | ∼16.66h |

*Table 5.* Memory requirements comparison

| Memory requirements | OOD features | OOD images | Total |
|---|---|---|---|
| BOOD | ∼7.32MB | ∼11.7G | ∼11.7G |
| DreamOOD | ∼2.9G | ∼11.67G | ∼14.57G |

ID features proximally distributed to the anchor.

**The effect of $\beta$.** From Figure 6 (middle), we can conclude that empirical evidence suggests optimal performance is achieved with moderate regularization weighting $\beta = 2.5$ , as excessive OOD regularization can compromise OOD detection efficiency.

**The effect of $K$.** We analyze the effect of maximum iteration number $K$ in Figure 6 (right). We vary $K \in \{5, 50, 100, 200, 400\}$ and found that a relatively large max iteration number $K$ to ensure comprehensive boundary crossing for most features. While increased iterations do affect computational overhead in boundary identification, the impact remains manageable.

### 4.3.3. COMPUTATIONAL COST AND MEMORY REQUIREMENTS

In this section, we conducted a comparative study of computational efficiency between BOOD and DreamOOD (Du et al., 2023). We specifically focus on four key processes:

(1) the building of latent space, (2) OOD features synthesizing, (3) the OOD image generation and (4) regularization of OOD detection model. To provide quantitative evidence, we present a detailed comparison of computational requirements between BOOD and DreamOOD (Du et al., 2023) in table 4. We also summarize the storage memory requirements of BOOD and DreamOOD (Du et al., 2023) on CIFAR-100 in table 5. Our empirical evaluation reveals that the differences between these approaches are not statistically significant. Thus, our proposed framework is not time consuming or has strict memory requirements.

## 5. Related Work

**OOD detection.** OOD detection has experienced a notable increase in research attention, as evidenced by numerous studies (Tajwar et al., 2021; Fort et al., 2021; Elflein et al., 2021; Fang et al., 2022; Yang et al., 2022; 2024a). A branch of research approach to addressing the OOD detection problem through designing scoring mechanisms, such as Bayesian approach (Gal & Ghahramani, 2016; Lakshminarayanan et al., 2017; Malinin & Gales, 2018; Osawa et al., 2019), energy-based approach (Liu et al., 2020; Lin et al., 2021; Choi et al., 2023) and distance based methods (Abati et al., 2019; Ren et al., 2021; Zaeemzadeh et al., 2021; Ming et al., 2023). Katz-Samuels et al. (2022) and Du et al. (2024) propose methods harnessing unlabeled data to enhance OOD detection, illustrating possible directions

to leverage wild data which is considered abundant. Most of these works need auxiliary datasets for regularization. VOS (Du et al., 2022) and NPOS (Tao et al., 2023a) propose methodologies for generating outlier data in the feature space, DreamOOD (Du et al., 2023) and SONA (Yoon et al., 2025) synthesizes OOD images in the pixel space. Compared to DreamOOD (Du et al., 2023), NPOS (Tao et al., 2023a) and SONA (Yoon et al., 2025) which samples features with Gaussian-based strategies or in the nuisance region, BOOD synthesizes features located around the decision boundaries, providing high-quality information to the OOD detection model. Additionally, compared to Chen et al. (2020) and Pei et al. (2022) who train the OOD discriminator harnessing the OOD samples at the ID-OOD boundary at feature level, our framework locate the distribution boundary more explicitly and can generate image-level OOD images.

**Diffusion-model-based data augmentation.** The field of data augmentation with diffusion models attracts various attention (Tao et al., 2023b; Zhu et al., 2024; Ding et al., 2024; Yeo et al., 2024). One line of work performed image generation with semantic guidance. Dunlap et al. (2023) proposes to caption the images of the given dataset and leverage the large language model (LLM) to summarize the captions, thus generating augmented images with the text-to-image model. Li et al. (2024) generated augmented images with the guidance of captions and textual labels, which are generated from the image decoder and image labels. A branch of research proposed perturbation-based approaches to synthesize augmented images (Shivashankar & Miller, 2023; Fu et al., 2024). Zhang et al. (2023) perturbed the CLIP (Radford et al., 2021)-encoded feature embeddings, guided the perturbed features by class name token features, and finally decoded it with diffusion model. Our framework BOOD creates an image feature space aligning with the class token embeddings encoded by CLIP (Radford et al., 2021). It proposes a perturbation strategy to generate informative OOD features that are located around the decision boundary.

## 6. Conclusion

In this paper, we propose an innovative methodology BOOD that generates effective decision boundary-based OOD images via diffusion models. BOOD provides two key methodologies in identifying the ID boundary data and synthesizing OOD features. BOOD proves that generating OOD images located around the decision boundaries is effective in helping the detection model to form precise ID-OOD decision boundaries, thus delineating a novel trajectory for synthesizing OOD features within this domain of study. The empirical result demonstrates that the generated boundary-based outlier images are high-quality and informative, resulting in a remarkable performance on popular OOD detection benchmarks.

## 7. Limitations

Although BOOD achieves excellent performance on common benchmarks, it still has some shortcomings. For some datasets that has **large domain discrepancy** from the diffusion model's training distribution, BOOD's performance might be affected by the diffusion model's capability. The classification error for unseen outlier features in Section 3.2.2 might result in deviations in determining whether a perturbed feature has crossed the decision boundary, leading to generating low-quality OOD features. Besides, it might be difficult for tuning the hyper parameters. In the future study, designing a automatic adaptive method for tuning $c$ might improve the performance and reduce the training time.

## Acknowledgment

This research is supported by the National Natural Science Foundation of China (No. 62306046, 62422606, 62201484). We would also like to thank ICML anonymous reviewers for helpful feedback.

## Impact Statement

This paper presents work whose goal is to advance the field of Machine Learning, specifically in the OOD data generation and detection field. There are many potential societal consequences of our work, none which we feel must be specifically highlighted here.

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

## A. Datasets details

**ImageNet-100.** For IMAGENET-100, we choose the following 100 classes from IMAGENET-1K following DreamOOD's (Du et al., 2023) setting: n01498041, n01514859, n01582220, n01608432, n01616318, n01687978, n01776313, n01806567, n01833805, n01882714, n01910747, n01944390, n01985128, n02007558, n02071294, n02085620, n02114855, n02123045, n02128385, n02129165, n02129604, n02165456, n02190166, n02219486, n02226429, n02279972, n02317335, n02326432, n02342885, n02363005, n02391049, n02395406, n02403003, n02422699, n02442845, n02444819, n02480855, n02510455, n02640242, n02672831, n02687172, n02701002, n02730930, n02769748, n02782093, n02787622, n02793495, n02799071, n02802426, n02814860, n02840245, n02906734, n02948072, n02980441, n02999410, n03014705, n03028079, n03032252, n03125729, n03160309, n03179701, n03220513, n03249569, n03291819, n03384352, n03388043, n03450230, n03481172, n03594734, n03594945, n03627232, n03642806, n03649909, n03661043, n03676483, n03724870, n03733281, n03759954, n03761084, n03773504, n03804744, n03916031, n03938244, n04004767, n04026417, n04090263, n04133789, n04153751, n04296562, n04330267, n04371774, n04404412, n04465501, n04485082, n04507155, n04536866, n04579432, n04606251, n07714990, n07745940.

## B. Additional visualization of the generated images

In this section, we provide additional visualizations of generated images.

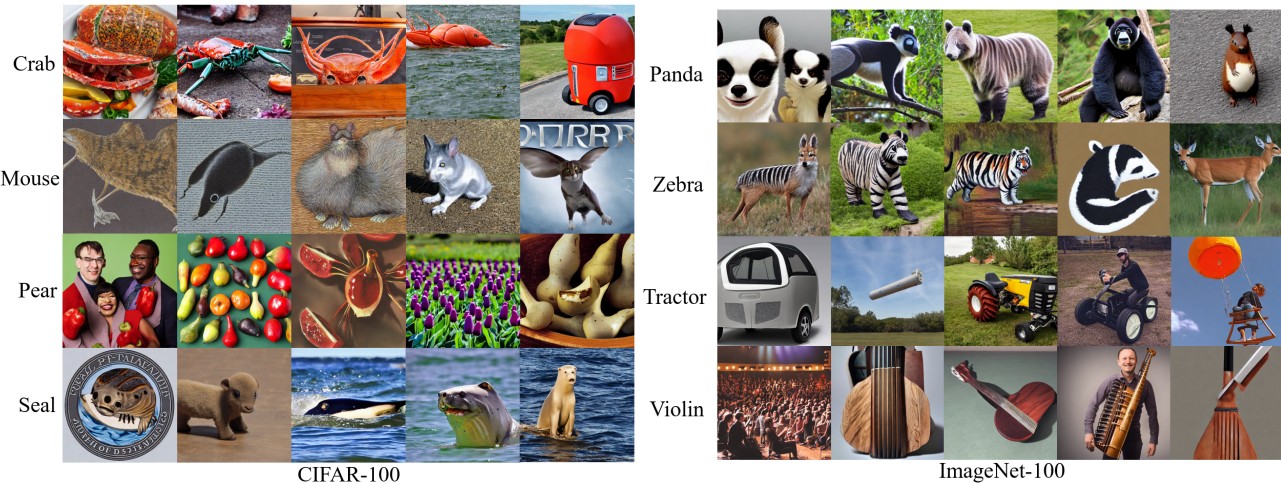

*Figure 7.* **Left**: OOD images generated for CIFAR-100. **Right**: OOD images generated for IMAGENET-100.

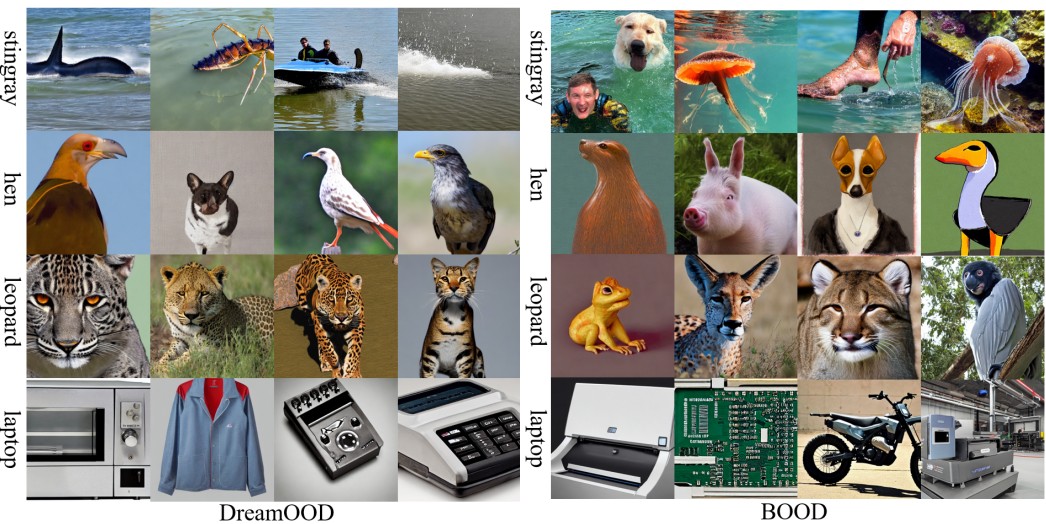

*Figure 8.* **Left**: OOD images generated with DreamOOD (Du et al., 2023). **Right**: OOD images generated with BOOD.

## C. Comparison between perturbation methods

To gain a deeper insight of the effectiveness of our strategy, we provide additional ablation studies (see table 6) on the different perturbation strategies in this section, including (1) adding Gaussian noises to the latent features, (2) displacing features away from class centroids and (3) BOOD's perturbation strategy. The results illustrates that our perturbation strategies are solid.

*Table 6.* Comparison of BOOD with different perturbation methods

| Method | FPR95 ↓ | AUROC ↑ |
|--------|---------|---------|
| (1)    | 18.99   | 95.04   |
| (2)    | 40.51   | 91.63   |
| BOOD   | 10.67   | 97.42   |

## D. BOOD's performance on NearOOD datasets

We conduct the experiment on two selected NearOOD datasets: NINCO (Bitterwolf et al., 2023) and SSB-HARD (Vaze et al., 2022). Below are performance results on NearOOD using IMAGENET-100 as the ID dataset:

*Table 7.* Performance on NearOOD datasets using ImageNet-100 as the ID dataset

| Method | NINCO | | SSB-HARD | | Average | |
|--------|---------|---------|---------|---------|---------|---------|
|        | FPR95 ↓ | AUROC ↑ | FPR95 ↓ | AUROC ↑ | FPR95 ↓ | AUROC ↑ |
| DreamOOD (Du et al., 2023) | 57.08 | 89.51 | 77.29 | 74.77 | 67.19 | 82.14 |
| BOOD | 54.37 | 89.82 | 75.48 | 80.82 | 64.93 | 85.32 |

From the table above we can see that in the NearOOD dataset, BOOD shows **better performance** than DreamOOD (Du et al., 2023), thus indicating that BOOD can handle more complex OOD datasets.

## E. Architectures of model

For code reproducibility, we introduce our model selection for image encoder(Sec 3.1) and OOD regularization model (Sec 3.3) here: we choose a standard ResNet-34 (He et al., 2016) for both of them, with the final linear transformation layer changed to $512 \rightarrow 768$ for image encoder (aligns with class token embeddings).

We also provide comparison between BOOD's performance on ResNet-18 (He et al., 2016), ResNet-34 (He et al., 2016) and ResNet-50 (He et al., 2016) using CIFAR-100 as ID dataset:

*Table 8.* Performance comparison across different ResNet backbones

| Backbone | FPR95 ↓ | AUROC ↑ | ID ACC ↑ |
|----------|---------|---------|----------|
| ResNet-18 (He et al., 2016) | 10.83 | 97.74 | 78.11 |
| ResNet-34 (He et al., 2016) | 10.67 | 97.42 | 78.03 |
| ResNet-50 (He et al., 2016) | 11.23 | 96.89 | 79.68 |

From the table above we can see that the selection of backbone architecture will **not significantly** affect the performance of BOOD.

