# OpenReview forum: "BOOD: Boundary-based Out-Of-Distribution Data Generation"
_ICML.cc/2025/Conference — ICML 2025 poster_

### Official Review · Reviewer_sEzZ · 2025-03-07

**Overall Recommendation:** 3

**Summary:**

This paper focuses on addressing the OOD detection task by synthesizing OOD samples. To generate plausible OOD samples, samples near the OOD boundary are first selected and then perturbed along the direction of gradient ascent until their predicted labels change. Finally, a diffusion model is applied to generate OOD samples from these perturbed features, which are then used to train an OOD classifier. Experiments on various datasets demonstrate that this method outperforms existing approaches.

**Claims And Evidence:**

The claims are clear; however, the method has inherent limitations that may push an in-distribution (ID) class toward another ID class.

**Essential References Not Discussed:**

All relevant works have been appropriately cited in the paper, and there are no crucial studies or findings that have been overlooked.

**Experimental Designs Or Analyses:**

Yes, I have checked the experimental designs. The paper uses CIFAR-100 and ImageNet-100 as the ID datasets, which are consistent with the DreamOOD settings. The experimental setup seems sound and appropriately chosen for the task at hand.

**Methods And Evaluation Criteria:**

The proposed methods are clear; however, there are certain limitations. The performance on the benchmark datasets is acceptable.

**Other Comments Or Suggestions:**

I have no additional comments or suggestions.

**Other Strengths And Weaknesses:**

Strengths:
1.	This paper proposes a novel boundary-based OOD data generation method that leverages a diffusion model to identify ID data closest to the decision boundary and applies an outlier feature synthesis strategy to generate images near the decision boundary. This approach provides high-quality and information-rich features for OOD detection.
2.	In terms of performance, the gain on CIFAR-100 as the ID is significant compared to existing methods.
3.	The writing and expression of the paper are clear. The images generated under boundary conditions are reasonable.
Weaknesses:
1.	How can we ensure that during the generation from 0 to c, the original ID sample of class y is not mistakenly perturbed into another ID class? Simply controlling the step size α and number of steps c may make the model overly sensitive to hyperparameters.
2.	I would like to understand whether the performance improvement over the DreamOOD method arises from generating OOD data with lower error rates, or from a higher number of OOD samples being situated closer to the decision boundary.
3.	I have noticed that 100,000 generated OOD images were used during training. Would reducing or increasing the number of images have an impact on the ID accuracy or OOD metrics?

**Questions For Authors:**

I have no further questions for the authors.

**Relation To Broader Scientific Literature:**

This paper, inspired by DreamOOD's use of nearest neighbor distance for OOD sample generation, employs gradient ascent to generate OOD samples, providing a more directed generation approach.

**Theoretical Claims:**

I have reviewed the theoretical claims, which primarily involve gradient ascent and related theoretical concepts. The limitations of the proposed method are already reflected in the theoretical claims.

---

> ### Author Rebuttal · Authors · 2025-03-31
>
> # Response to reviewer sEzZ
>
> We thank the reviewer for the feedback and constructive suggestions. Our response to the reviewer’s concerns is below:
>
> > Weakness 1: How can we ensure that during the generation from 0 to c, the original ID sample of class y is not mistakenly perturbed into another ID class? Simply controlling the step size α and number of steps c may make the model overly sensitive to hyperparameters.
>
> We appreciate this important inquiry regarding hyperparameters. We would like to emphasize that OOD features are the features that distributed between classes, so they are **ambiguous and likely to confuse the classifier**. But if BOOD generates features that are distributed in the ID area, the final OOD image will be considered as **noisy data** and **be harmful to the model performance**. While we don't have an explicit way to measure whether image features have been mistakenly perturbed into another ID class, **by setting relatively small $\alpha$ and $c$ values**, we can guarantee that the generated OOD features will located around the decision boundaries.  Figure 5 shows that the performance of OOD detection rises firstly and latter decrases as $c$ increases, which illustrates **our control of $c$ are effective and most of the generated OOD images are effective**. In the next step, we are working on filtering mechanism to rule out the potential mistakenly perturbed features.
>
> > Weakness 2: I would like to understand whether the performance improvement over the DreamOOD method arises from generating OOD data with lower error rates, or from a higher number of OOD samples being situated closer to the decision boundary.
>
> Thank you for the interesting point. Would you clarify the meaning of "**lower error rates**" in the context?
>
> Assume the "lower error rates" indicates **the rate of mistakenly generated ID images in the generated OOD data**. Since OOD data are the samples that are not distributed in the input data distributions, thus they are hard to be classified by the classifier and difficult to decide whether they are OOD samples. Our method intend to find the boundaries of ID data and perturb ID features which are cloestest to the boundary to the area between classes and close to the decision boundaries in the latent space, thus it's diffiucult to quantify the error rate of the generated OOD data.
>
> We also conduct the experiment comparing the feature's average distance to their closest decision boundaries between DreamOOD[1] and our method:
>
> | Average distance | ImageNet-100 | CIFAR-100 |
> |:--:|:--:|:--:|
> |DreamOOD | 3.70 | 5.23 |
> |BOOD| 2.29 | 4.01 |
>
>
> The results show that BOOD's generated features are **closer to the closest decision boundaries**, resulting in the improvement of performance in OOD detection task.
>
> > Weakness 3: I have noticed that 100,000 generated OOD images were used during training. Would reducing or increasing the number of images have an impact on the ID accuracy or OOD metrics?
>
> Great point! Here we list the performance statistics of BOOD and DreamOOD[1] training on different number of OOD images (use CIFAR-100 as ID dataset):
>
> | Total OOD images | FPR95 $\downarrow$ | AUROC $\uparrow$ | ID ACC $\uparrow$ |
> |:--:|:--:|:--:|:--:|
> | DreamOOD-10k | 60.23 | 81.84 | 65.39 |
> | DreamOOD-50k | 48.66 | 85.71 | 72.95 |
> | DreamOOD-100k | 40.31 | 90.15 | 78.94 |
> | BOOD-10k | 25.21 | 93.63 | 65.14 |
> | BOOD-50k | 15.83 | 96.1 | 73.18 |
> | BOOD-100k | 12.47 | 97.34 | 78.17 |
>
> From the table above we can find that **as the number of OOD training images increases**, both BOOD and DreamOOD's[1] performance on OOD detection and ID classification **increases**.
>
> We hope we have responded to all your concerns. If you have further questions, we are pleased to discuss with you. Thank you again for taking the time to read our response and your constructive feedback!
>
> [1] Xuefeng Du, Yiyou Sun, Xiaojin Zhu and Yixuan Li. Dream the impossible-Outlier imagination with diffusion models. NeurIPS, 2023

---

### Official Review · Reviewer_Mr32 · 2025-03-09

**Overall Recommendation:** 4

**Summary:**

This paper studies the problem of out-of-distribution (OOD) detection for image tasks. The authors leverage text-to-image latent diffusion models to synthesize OOD images that are used to train the binary OOD detector. In doing so, they follow a three-step strategy: 1) Identifying the ID samples that are closest to the decision boundary by measuring the number of steps (k) required to perturb the ID feature along the gradient ascent direction until the model's prediction changes (this number of steps serves as a proxy for the shortest distance between the InD sample and the decision boundary); 2) perturbing the identified features from step 1 to get OOD embeddings, and 3) using OOD embeddings from step 2 to generate OOD images using the diffusion model. Experiments on CIFAR-100 and ImageNet-100 as ID show improvements in performance from previous scoring-based methods as well as synthesis-based methods.

**Claims And Evidence:**

Yes, most claims are supported by evidence.

However, some parts could be clearer. For example, in the abstract, the authors state that “…BOOD provides a more efficient strategy for synthesizing informative OOD features…”. However, there is no clear evidence showing that the method is truly efficient. Although the authors compare computation and memory usage in Tables 5 and 6, the differences are not significant, and in many cases, BOOD uses more computation than DreamOOD. I also believe the comparison should include more baselines, not just one. It might be better to either remove this claim or rephrase it more modestly.

Additionally, in Section 4.3.1, the authors claim that “Employing a large c may force the feature to step into the ID region.” But is there evidence to support this, apart from the performance degradation? Would it be possible to provide an experiment (like a visual example) showing that the feature indeed returns to the ID region? For instance, it would have been useful to see this in Figure 3 (left) with more steps, where the image might begin to resemble the ID image.

**Essential References Not Discussed:**

While the paper discusses most of the relevant works, there is a recent synthesis-based method that was not mentioned, let alone compared against, in the paper. Specifically, the work titled Diffusion-based Semantic Outlier Generation via Nuisance Awareness for Out-of-Distribution Detection by Suhee Yoon et al. (2024) (https://arxiv.org/abs/2408.14841) presents an interesting approach that leverages diffusion models for OOD detection. Including a discussion of this work would have enhanced the paper's context and provided a valuable comparison.

There are some other important studies that could have been included to provide a more comprehensive context. Recent works, for instance, have explored the use of unlabeled wild data to enhance OOD detection, achieving state-of-the-art results on datasets similar to those used in this paper. For example, Du et al. (2024) achieve an FPR95 of 0.07 on the CIFAR100-SVHN ID-OOD pair, compared to 5.42 in this paper. Similarly, on the CIFAR100-PLACES365 ID-OOD pair, Du et al. (2024) report an FPR95 of 3.53, while this paper has an FPR95 of 40.55. Some of them are:

•	Xuefeng Du, Zhen Fang, Ilias Diakonikolas, and Yixuan Li. "How does unlabeled data provably help out-of-distribution detection?" ICLR, 2024.

•	Julian Katz-Samuels, Julia Nakhleh, Robert Nowak, and Yixuan Li. "Training OOD detectors in their natural habitats." ICML, 2022.

While these studies use unlabeled data, whereas this paper focuses on a synthesis-based method, it would have been valuable to discuss the literature on unlabeled data and its impact on OOD detection, as it provides important context and alternative approaches to the problem.

**Experimental Designs Or Analyses:**

Yes, I checked the algorithm and experimental designs, and they look reasonable to me.

**Methods And Evaluation Criteria:**

Yes.

**Other Comments Or Suggestions:**

Minor: in the related work section, use ‘citet’ instead of ‘citep’ for “…guidance. (Dunlap et al., 2023) proposes to caption the images…” and “…model. (Li et al., 2024) generated augmented images with the guidance…”.

**Other Strengths And Weaknesses:**

**Strengths:**

The paper is well-motivated, well-written, and easy to follow. The methodology is clearly explained, with illustrative examples provided in the figures, which helped in understanding better.

Generally, I finding the algorithm design decisions quite intuitive and systematic. The method is simple yet effective.

**Weaknesses:**

The third step of the method (the OOD image generation) is based on the approach in Du et al. (2023), while the regularization technique is derived from VOS (Du et al., 2022). These elements are largely borrowed from prior works and integrated into the BOOD framework. This raises concerns about the novelty of the method.

The authors did not provide their code to reproduce the results, which raises concerns about the transparency of the findings.

**Questions For Authors:**

Why is regularization used in the OOD detection model in Section 3.3? What is the significance of this regularization, and what is the underlying intuition? It would be helpful to mention this in this paragraph. Moreover, why does setting such a large value of $\beta=2.5$ help; what could be a potential rationale?

**Relation To Broader Scientific Literature:**

The paper addresses the OOD detection problem, which is a well-established problem in the machine learning literature. Most previous works have focused on using scoring methods (e.g., MSP, DICE, Energy, etc.) to tackle the OOD detection problem. More recent works have shifted towards generating OOD samples using generative models, such as GANs and diffusion models. This work adopts the latter approach, utilizing diffusion models to generate OOD samples and then trains a binary OOD detector using both ID samples and the synthesized OOD samples.

**Theoretical Claims:**

N/A

---

> ### Author Rebuttal · Authors · 2025-03-31
>
> # Response to reviewer Mr32
>
> We appreciate the review for providing valuable advice. Below are our responses:
>
> > Claims and Evidence 1: there is no clear evidence showing that the method is truly efficient
>
> We apologize for the inaccurate expression in the abstract: BOOD provides a more **training efficient** strategy for synthesizing informative OOD features. We will fix this part in the updated manuscript, thank you again for pointinng this out.
>
> > Claims and Evidence 2: need to provide a visual example showing that the feature indeed returns to the ID region.
>
> Thank you for your advice. Please check **Figure 2** in our paper, which shows the perturbation process of two features from one ID class to another ID class: one from **Tiger** to **Fish**, another one from **Streetcar** to **House**. We will also include more visual examples illustrating the perturbation process.
>
> > Some essential References Not Discussed.
>
> We extend our gratitude to the reviewer for the recommendation of essential references. We will update the corresponding part in the related works of paper in the camera-ready version.
>
> > Weakness 1: The third step of the method (the OOD image generation) is based on the approach in Du et al. (2023), while the regularization technique is derived from VOS (Du et al., 2022). These elements are largely borrowed from prior works and integrated into the BOOD framework. This raises concerns about the novelty of the method.
>
> We thank the reviewer for the concern regarding the novelty. We would like to emphasize that **our framework mainly focus on the generation of imformative OOD features** in the latent space. **To guarantee the fairness in comparison with the previous SOTA method DreamOOD in synthesis-based OOD detection frameworks**, we follow the same regularization function as them. After generating the OOD dataset, the regularization function can be substitute by other common functions to train the OOD detection model.
>
> > Weakness 2: Code not provided.
>
> Thanks for concern regarding the code. We will release our code, data and model once the paper is accepted.
>
> > Comments: In the related work section, use ‘citet’ instead of ‘citep’ for “…guidance. (Dunlap et al., 2023) proposes to caption the images…” and “…model. (Li et al., 2024) generated augmented images with the guidance…”.
>
> Thanks for pointing out the minor mistakes, we will update them in the revised manuscript.
>
> > Question: Why is regularization used in the OOD detection model in Section 3.3? What is the significance of this regularization, and what is the underlying intuition? It would be helpful to mention this in this paragraph. Moreover, why does setting such a large value of $\beta$ help; what could be a potential rationale?
>
> The significance for regularization the OOD detection model is to keep model's ability of performing visual recognition task while regularizing the model to identify the generated OOD images. Equation (7) shapes the uncertainty surface, which predicts high probability for ID data and low probability for OOD data. Compared to other regularization method such as Energy-bounded[1], Equation (7) is **hyperparameter-free so easier to be implemented.**
>
> Our analysis suggest setting a relatively larger regularization weighting $\beta$ to ensure better performance (Figure 6 middle). The potential reationale could be: using a relatively larger $\beta$ will help to **force the model to push OOD samples away** thus enhance the model's ablility for distinguishing ID and OOD input. However, if the $\beta$ is too large, the model will be **over-regularized** and the performance for OOD detection will decrease.
>
> We sincerely thank you for reviewing our response and your constructive feedback. If any aspects still need clarification, we are happy to discuss them further.
>
> [1] Weitang Liu, Xiaoyun Wang, John Owens and Yixuan Li. Energy-based out-of-distribution detection. NeurIPS, 2020

---

> > ### Comment · Reviewer_Mr32 · 2025-04-04
> >
> > I appreciate the authors for their rebuttal and will keep my rating for acceptance.

---

> > > ### Author Response · Authors · 2025-04-04
> > >
> > > We express our gratitude to the reviewer again for your valuable time and constructive feedback!

---

### Official Review · Reviewer_Smhu · 2025-03-14

**Overall Recommendation:** 4

**Summary:**

This paper introduces a framework called Boundary-based Out-Of-Distribution data generation (BOOD). BOOD synthesizes high-quality OOD features and generates outlier images using diffusion models.

The BOOD framework learns a text-conditioned latent feature space from the ID dataset, selects ID features closest to the decision boundary, and perturbs them to cross the decision boundary to form OOD features.  These synthetic OOD features are then decoded into images in pixel space by a diffusion model.

The authors claim that BOOD provides a more efficient strategy for synthesizing informative OOD features, facilitating clearer distinctions between ID and OOD data. Experimental results on common benchmarks demonstrate that BOOD surpasses the state-of-the-art method significantly.

**Claims And Evidence:**

The claims made in the submission are generally well-supported by clear and convincing evidence. The authors have provided a thorough evaluation of their proposed method, BOOD, and compared it with several state-of-the-art approaches. The experimental results on common benchmarks demonstrate that BOOD surpasses the state-of-the-art method significantly.

Specifically, the main claims are supported by the following evidence:

* **Claim:** BOOD surpasses the state-of-the-art method significantly.

    * **Evidence:** The experimental results in Table 1 and Table 2 show that BOOD outperforms other methods on both CIFAR-100 and IMAGENET-100 datasets.

* **Claim:** BOOD provides a more efficient strategy for synthesizing informative OOD features, facilitating clearer distinctions between ID and OOD data.

    * **Evidence:** The ablation studies in Section 4.3.1 and the visualization of generated images support this claim.

* **Claim:** The proposed framework is not time-consuming or has strict memory requirements.

    * **Evidence:** The computational cost comparison in Table 5 and the memory requirements comparison in Table 6 demonstrate that BOOD's computational and memory demands are comparable to those of DreamOOD, a state-of-the-art method.

However, it is important to note that BOOD had many hyperparameters that affect the results significantly such as the step size and the number of perturbations steps and require some tuning.

**Essential References Not Discussed:**

The key contribution of this paper relies primarily on identifying the in-distribution (ID) features located near the decision boundary and subsequently perturbing these features away from that boundary to generate out-of-distribution (OOD) data. To provide context for this approach, it would be valuable to include references [1,2], as they propose methods for analyzing and approximating the distance to the decision boundary using adversarial attack concepts similar to those employed in this paper. These works offer complementary perspectives on understanding and characterizing decision boundaries in deep neural networks.


### References:

1- Mickisch, David, et al. Understanding the Decision Boundary of Deep Neural Networks: An Empirical Study. arXiv:2002.01810, arXiv, 5 Feb. 2020. arXiv.org, https://doi.org/10.48550/arXiv.2002.01810.

2- Yousefzadeh, Roozbeh, and Dianne P. O’Leary. “Deep Learning Interpretation: Flip Points and Homotopy Methods.” Proceedings of The First Mathematical and Scientific Machine Learning Conference, PMLR, 2020, pp. 1–26. proceedings.mlr.press, https://proceedings.mlr.press/v107/yousefzadeh20a.html.

**Experimental Designs Or Analyses:**

The authors have taken reasonable steps to ensure the reliability and robustness of their findings. Here's a breakdown of the key aspects:

**1. Ablation Studies:**

* The authors conduct ablation studies to analyze the contribution of different components of their proposed BOOD framework.

* Specifically, they ablate the effect of boundary identification and feature perturbation.

* By comparing the performance of the full BOOD framework with variants where these components are removed or replaced, they demonstrate the importance of each component for achieving the best results.

* This ablation analysis helps to validate the design choices made in the BOOD framework.

**2. Hyperparameter Analysis:**

* The authors perform a hyperparameter sensitivity analysis to evaluate the impact of different hyperparameter settings on the performance of BOOD.

* They analyze the effect of step size, perturbation steps after crossing the boundary, pruning rate, OOD regularization weighting, and maximum perturbation steps.

* By varying these hyperparameters and observing the resulting changes in performance, they gain insights into the optimal settings for BOOD and the sensitivity of the method to these parameters.

* This analysis helps to ensure that the reported results are not due to a specific choice of hyperparameters and provides guidance for applying BOOD in different settings.

**3. Comparison with State-of-the-Art Methods:**

* The authors compare the performance of BOOD with several state-of-the-art OOD detection methods on benchmark datasets.

* This comparison allows them to demonstrate the superiority of BOOD over existing methods and provides evidence for the effectiveness of their proposed approach.

* The use of standard benchmark datasets and evaluation metrics ensures that the comparison is fair and objective.


Overall, the experimental designs and analyses in the paper are well-structured, comprehensive, and appropriate for evaluating the proposed BOOD framework. The authors have carefully considered various factors that could affect the validity of their results and have taken steps to address them through ablation studies, hyperparameter analysis, and comparison with state-of-the-art methods.

**Methods And Evaluation Criteria:**

The proposed methods and evaluation criteria in this paper are generally well-suited for the problem of Out-Of-Distribution (OOD) detection.

* BOOD's approach of synthesizing OOD features by perturbing ID features near the decision boundary is a sound method for generating informative outliers.

* The use of diffusion models to generate human-compatible outlier images is a reasonable choice, given their strong generative capabilities

* The alignment of the image feature space with the diffusion-model-input space is a crucial step to ensure compatibility and effectiveness of the generated OOD data as done in DreamOOD.

**Evaluation Criteria:**

* The paper employs widely used benchmark datasets for OOD detection, such as CIFAR-100 and IMAGENET, which allows for comparison with other state-of-the-art methods.

* The evaluation metrics used, including False Positive Rate at 95% True Positive Rate (FPR95) and Area Under the Receiver Operating Characteristic Curve (AUROC), are standard metrics for evaluating OOD detection performance.

* The inclusion of in-distribution classification accuracy (ID ACC) as an evaluation metric is important to ensure that the OOD detection method does not compromise the model's performance on in-distribution data.

Overall, the proposed methods and evaluation criteria are appropriate and well-justified for addressing the problem of OOD detection.

However, it would be beneficial to report OOD performance on ImageNet-200 and CIFAR-10 and also to add distinction in test set between Near and Far OOD.

**Other Comments Or Suggestions:**

NA

**Other Strengths And Weaknesses:**

**Strengths:**

* **Originality:**
    * The paper introduces a novel framework, BOOD, for generating OOD data.
    * BOOD's originality lies in its explicit focus on generating OOD images located around the decision boundaries between classes.
* **Significance:**
    * The proposed BOOD framework offers a promising approach to improve OOD detection by generating informative OOD data.
    * The experimental results demonstrate that BOOD outperforms state-of-the-art methods on benchmark datasets, highlighting the significance of the contribution.
* **Clarity:**
    * The paper is generally well-written and easy to follow.
    * The problem is clearly defined, and the proposed approach is well-motivated.

**Weaknesses:**
* **Hyperparameter Sensitivity:**
    * The performance of BOOD depends on several hyperparameters, such as step size, perturbation steps, pruning rate, and OOD regularization weighting.
    * Although the authors conduct a hyperparameter sensitivity analysis, the process of tuning these parameters for new datasets or applications may be challenging.
* **Generalization to More Complex Datasets:**
    * The experiments are conducted on CIFAR-100 and IMAGENET-100.
    * While these are standard benchmark datasets, it is not clear how well BOOD would generalize to more complex and datasets or to NearOOD test sets as the one in the paper are considered FarOOD.

**Questions For Authors:**

1 -  The authors mention that they "employ a class embedding alignment strategy during the image encoder training following Du et al. (2023)". It would be helpful to understand why this strategy is important for generating OOD images. Does this alignment ensure compatibility between the feature space and the diffusion model?

2- The authors use an adversarial perturbation strategy to identify ID boundary features. How sensitive is this boundary identification process to the choice of hyperparameters, such as the step size α and the maximum iteration number K? How do these parameters influence the accuracy of identifying features closest to the decision boundary?

3- The authors compare BOOD with several state-of-the-art OOD detection methods. Could they discuss how BOOD's performance might be affected by the choice of the backbone architecture or the diffusion model used for image generation?

**Relation To Broader Scientific Literature:**

The key contributions of the paper are related to the broader scientific literature in the following ways:

* **Out-of-Distribution (OOD) Detection:** The paper addresses the problem of OOD detection, which is a well-established area of research in machine learning.

* The goal of OOD detection is to identify inputs that come from a different distribution than the one the model was trained on.

* **Data Augmentation with Diffusion Models:** The paper leverages the power of diffusion models for data augmentation, which is a rapidly growing area of research.

* Diffusion models have shown remarkable success in generating high-quality and diverse images.

* The paper's approach is related to other works that have used diffusion models for data augmentation, including those that perform image generation with semantic guidance and those that use perturbation-based approaches to synthesize augmented images.

* **Synthesizing OOD Data:** A core contribution of the paper is the development of a novel framework, BOOD, for synthesizing OOD data.

* This contribution is closely related to prior work that has explored the use of auxiliary outlier datasets to improve OOD detection.

* **Explicitly Generating OOD Images near Decision Boundaries:** BOOD's innovation lies in its ability to generate image-level OOD data located around the decision boundaries between classes.

**Theoretical Claims:**

While the paper presents a novel framework and demonstrates strong empirical results, it doesn't contain significant theoretical claims that require in-depth proof verification. The core contributions are algorithmic and experimental, focusing on a new way to generate OOD data.

---

> ### Author Rebuttal · Authors · 2025-03-31
>
> # Response to reviewer Smhu
>
> > Essential References Not Discussed.
>
> Thanks for recommendation for essential references. We will add them into our camera-ready paper.
> > Weakness 1: the process of tuning hyperparameters may be challenging.
>
> We appreciate the reviewer for the meaningful concerning. While fine tuning the parameter $c$ and $\alpha$ may be somehow time-consuming, we would like to emphasize that $r$, $\beta$ and $K$'s effect are **not remarkable** (Figure 6): the difference in AUROC are within 0.5% when $r < 10$ and within 0.4% for $\beta \in \{1.5, 2, 2.5, 3\}$, the performance will not change if $K > 50$. Thus, our suggestions for tuning the hyperparameters can make the process of tuning less difficult. We are working on a possible automatic adaptive method to adjust the additional pertuabation steps $c$ in the next step, aiming to reduce the tuning time.
> > Weakness 2: BOOD's performance on ImageNet-200, CIFAR-10 and NearOOD datasets.
>
> Since our baseline method DreamOOD[1] does not include the performance on ImageNet-200 and CIFAR-10, we do not choose them as the ID dataset to guarantee comparison fairness. Due to the limited author response time, we cannot present the result of these two dataset, but we will include them in our camera-ready paper.
>
> We summerize the experiment results on two selected NearOOD datasets **NINCO and SSB-hard** as below:
>
> ||NINCO||SSB-hard||Average||
> |:-:|:-:|:-:|:-:|:-:|:-:|:-:|
> |Method|FPR95 &#8595;|AUROC &#8593;|FPR95 &#8595;|AUROC &#8593;|FPR95 &#8595;| AUROC &#8593;|
> |DreamOOD[1]|57.08|89.51|77.29|74.77|67.19| 82.14|
> |BOOD|54.37|89.82|75.48|80.82|64.93|85.32|
>
> From the table above we can see that in the NearOOD dataset, BOOD shows **better performance** than DreamOOD[1], thus indicating that BOOD can handle more complex OOD datasets.
>
> > Question 1: importance of class embedding alignment strategy, capability between feature space and the diffusion model.
>
> We appreciate the reviewer for the reasonable concern. We aim to create a latent space that is compatible for the input space of diffusion model, so the alignment between the image features and their corresponding class token embeddings is necessary. By training an image encoder with **Formula (2)**, we ensure that the image features can be decoded by the diffusion models. Please check **Section 3.1** in our paper for detailed explanation.
>
> > Question 2: hyperparameter $\alpha$ and $K$'s sensitivity on accuracy of identifying features closest to the decision boundary.
>
> While it's difficult to directly measure the accuracy, from our experiment on ablation of OOD feature synthesizing methodologies (Sec 4.3.1 and Table 3), we can find that **perturbing random ID features to the boundaries will significantly decrease the OOD detection performance** compared to using the ID features closest to the decision boundary. Therefore, we can measure the accuracy of identifying features closest to the decision boundary by the final OOD detection performance.
>
> Employing **a relatively small $\alpha$** facilitates a more nuanced differentiation between samples, and a large $\alpha$ may lead to **large discrepancy** between each iteration in adversarial perturbation, making the counting of distance to decision boundaries not accurate. Selecting **a relatively large max iteration number $K$** can ensure **comprehensive boundary crossing for most features**. While increased $K$ do affect computational overhead in boundary identification, **the impact remains manageable**. We provide detailed discussion about the sensitivity of the hyperparameters $\alpha$ and $K$ in Section 4.3.2, Figure 5 (left) and Figure 6 (right).
>
> > Question 3: backbone architecture and diffusion model's impact on BOOD's performance.
>
> We provide comparison between BOOD's performance on Resnet-18, Resnet-34 and Resnet-50 using CIFAR-100 as ID dataset:
>
> |Backbone|FPR95 &#8595;|AUROC &#8593;|ID ACC &#8593;|
> |:-:|:-:|:-:|:-:|
> |ResNet-18|10.83|97.74|78.11|
> |ResNet-34|10.67|97.42|78.03|
> |ResNet-50|11.23|96.89|79.68|
>
> From the table above we can see that the selection of backbone architecture will **not significantly affect the performance of BOOD.**
>
> For some datasets that has **large domain discrepancy** from the diffusion model's training distribution (Textures), BOOD's performance might be affected by the **diffusion model's capability**. But this **constraint is inherent to all methodologies utilizing diffusion-based generative data augmentation**. While future developments in generative modeling may address these limitations, **we emphasize that our primary goal is to leverage diffusion models to generate informative OOD images** thus increasing the OOD detection model's performance.
>
> We believe we have responded to all your concerns. If anything remains unclear, we would be pleased to have further discussions with you.
>
> [1] Xuefeng Du, Yiyou Sun, Xiaojin Zhu and Yixuan Li. Dream the impossible-Outlier imagination with diffusion models. NeurIPS, 2023

---

### Official Review · Reviewer_wxMm · 2025-03-14

**Overall Recommendation:** 4

**Summary:**

This paper introduces a framework for generating synthetic out-of-distribution (OOD) data by explicitly targeting decision boundaries in latent feature space. The proposed method, BOOD, employs an adversarial perturbation strategy to identify in-distribution (ID) features closest to the decision boundary and perturbs them along the gradient ascent direction to synthesize informative OOD features. These features are then decoded into human-compatible OOD images using a diffusion model. Experiments and ablation studies have presented to validate the efficiency of the framework.

**Claims And Evidence:**

yes

**Essential References Not Discussed:**

no

**Experimental Designs Or Analyses:**

yes

**Methods And Evaluation Criteria:**

yes

**Other Comments Or Suggestions:**

The quality of figures can be improved.

The size of tables in the appendix can be adjusted.

**Other Strengths And Weaknesses:**

Strengths:
1. The paper is well written and easy to follow.
2. The idea of exploring ood features in the boundary and using diffusion models to generate images from ood features is reasonable and interest.
3. The method achieves better results than the compared methods on most evaluated datasets.
4. The ability of generating image-level OOD samples would be beneficial to other domains of the community.

Weaknesses:
1. The compared methods are 2023 and before, how about the performance regarding to the most recent works?
2. The discussion of OOD methods misses most works on 2024.
3. The complexity analysis in appendix should be included in the main text, as the timing of diffusion-based methods is always an interested point for researchers in the community. Besides, the details about the complexity analysis are unclear. For instance, what the dataset used for Table 5, how about the timing when generating samples one by one? Does the memory requirements in Table 6 are for training? How about the inference?
4. There are many hyper-parameters. Although the paper provided curves of using different values, setting them for a new dataset is still not an easy job. This could be a disadvantage of this method for practical applications.


=================== post rebuttal ========================
After read the rebuttal and other reviews, I would like to raise my score from weak accept to accept.

**Questions For Authors:**

no

**Relation To Broader Scientific Literature:**

The paper could have broader impact on OOD detection, open-world pattern recognition methods, and image generation.

**Theoretical Claims:**

yes

---

> ### Author Rebuttal · Authors · 2025-03-31
>
> # Response to reviewer wxMm
>
> We thank the reviewer for the comments.
> > Weakness 1: The compared methods are 2023 and before, how about the performance regarding to the most recent works?
>
> Thank you for your concern regarding baseline methods. Please note that our work is based on using diffusion models to generate image-level OOD datasets. However, there are few work in this specific area. We provide comparison between BOOD and **a SOTA methods** FodFom[1] from ACMMM 2024, which also harnesses diffusion model to generate OOD images for enhancing OOD detection model. The results are summarized in the table below:
>
> |       | SVHN |  | LSUN-R |  | LSUN-C | | iSUN | | Textures |   | Places365 | |  Average |  |
> |:-:|:-:|:--:|:-:|:-:|:--:|:---:|:--:|:-:|:--:|:-:|:-:|:-:|:-:|:-:|
> | Method  | FPR95 &#8595; |  AUROC &#8593;  | FPR95 &#8595; | AUROC &#8593; | FPR95 &#8595;  |  AUROC &#8593; |FPR95 &#8595; |AUROC &#8593; |FPR95 &#8595; | AUROC &#8593; | FPR95 &#8595; | AUROC &#8593;   |  FPR95 &#8595; | AUROC &#8593; |
> |  FodFoM   | 33.19 | 94.02 | 28.24 | 95.09 | 26.79 | 95.04 | 33.06 | 94.45 |  35.44 | 93.38 | 42.30 | 90.68 | 33.17 | 93.78 |
> | BOOD | 5.42 | 98.43 | 0.10 | 99.94 | 2.06 | 99.25 | 0.22 | 99.91 | 5.1 | 98.74 | 40.55 | 90.76 | 8.91 | 97.84 |
>
> Compare to FodFom[1], **BOOD demonstrates superior performance**, illustrating its competitiveness.
>
> > Weakness 2: The discussion of OOD methods misses most works on 2024.
>
> Thanks for your question related to the discussion on recent OOD methods. We will include them into our camera-ready version.
>
> > Weakness 3: The complexity analysis in appendix should be included in the main text, as the timing of diffusion-based methods is always an interested point for researchers in the community. Besides, the details about the complexity analysis are unclear. For instance, what the dataset used for Table 5, how about the timing when generating samples one by one? Does the memory requirements in Table 6 are for training? How about the inference?
>
> Thanks for pointing out the this. We will include the detailed complexity analysis in the main text in our revised manuscript.
>
> Regarding the dataset used in Table 5, we use **CIFAR-100** as the ID dataset. It takes around **4** second to generate a single image in our framework on **Stable Diffusion v1.4** with **one NIVDIA L40S GPU**. And for Table 6, the memory requirements indicates the space needed for **storing the generated OOD images** used for training the final OOD detection model. There's no extra memory requirements during the inference time.
>
> > Weakness 4: There are many hyper-parameters. Although the paper provided curves of using different values, setting them for a new dataset is still not an easy job. This could be a disadvantage of this method for practical applications.
>
> We appreciate the reviewer for the meaningful concerning. While fine tuning the parameter $c$ and $\alpha$ may be somehow time-consuming, we would like to emphasize that $r$, $\beta$ and $K$'s effect are **not remarkable** (Figure 6): the difference in AUROC are within 0.5% when $r < 10$ and within 0.4% for $\beta \in \{1.5, 2, 2.5, 3\}$, the performance will not change if $K > 50$. Thus, our suggestions for tuning the hyperparameters can make the process of tuning less difficult. We are working on a possible automatic adaptive method to adjust the additional pertuabation steps $c$ in the next step, aiming to reduce the tuning time.
>
> > Weakness 5: The quality of figures can be improved. The size of tables in the appendix can be adjusted.
>
> Thank you for your advice, we will polish all the figures and tables in our camera-ready version of paper.
>
> We thank the reviewer for taking the time to read our response and the positive feedback again. If you have any further concerns, we are willing to have discussions with you.
>
> [1] Jiankang Chen, Ling Deng, Zhiyong Gan, Wei-Shi Zheng and Ruixuan Wang. "FodFoM: Fake Outlier Data by Foundation Models Creates Stronger Visual Out-of-Distribution Detector." ACMMM 2024.

---

### Decision · Program_Chairs · 2025-05-01

**Decision:**

Accept (poster)

**Comment:**

This paper introduces BOOD, a framework for synthesizing out-of-distribution (OOD) data by explicitly perturbing in-distribution (ID) features near decision boundaries in latent space and decoding them into images via diffusion models. The method aims to generate informative OOD samples to improve OOD detection performance. All reviewers (wxMm, Smhu, Mr32, sEzZ) acknowledged the paper’s merits, including technical soundness and empirical performance.

The authors addressed some concerns (e.g., adding FodFoM comparisons, complexity analysis), but some limitations remain relevant for future work. The camera-ready version could better contextualize these weaknesses (e.g., computational costs as inherent to diffusion models, hyperparameter tuning guidelines), if the paper is finally accepted.